# Cationic Azobenzenes as Light-Responsive Crosslinkers for Alginate-Based Supramolecular Hydrogels

**DOI:** 10.3390/polym16091233

**Published:** 2024-04-28

**Authors:** Miriam Di Martino, Lucia Sessa, Barbara Panunzi, Rosita Diana, Stefano Piotto, Simona Concilio

**Affiliations:** 1Department of Pharmacy, University of Salerno, Via Giovanni Paolo II, 132, 84084 Fisciano, Italy; midimartino@unisa.it (M.D.M.); piotto@unisa.it (S.P.); 2BIONAM Research Center for Biomaterials, University of Salerno, 84084 Fisciano, Italy; 3Department of Agriculture, University of Napoli Federico II, Via Università 100, 80055 Portici, Italy; barbara.panunzi@unina.it (B.P.); rosita.diana@unina.it (R.D.)

**Keywords:** ionic azobenzenes, photoisomerization, supramolecular hydrogels, sodium alginate, crosslinkers

## Abstract

Azobenzene photoswitches are fundamental components in contemporary approaches aimed at light-driven control of intelligent materials. Significant endeavors are directed towards enhancing the light-triggered reactivity of azobenzenes for such applications and obtaining water-soluble molecules able to act as crosslinkers in a hydrogel. Here, we report the rational design and the synthesis of azobenzene/alginate photoresponsive hydrogels endowed with fast reversible sol–gel transition. We started with the synthesis of three cationic azobenzenes (AZOs A, B, and C) and then incorporated them in sodium alginate (SA) to obtain photoresponsive supramolecular hydrogels (SMHGs). The photoresponsive properties of the azobenzenes were investigated by UV–Vis and ^1^H NMR spectroscopy. Upon irradiation with 365 nm UV light, the azobenzenes demonstrated efficient *trans*-to-*cis* isomerization, with complete isomerization occurring within seconds. The return to the *trans* form took several hours, with AZO C exhibiting the fastest return, possibly due to higher *trans* isomer stability. In the photoresponsive SMHGs, the minimum gelation concentration (MGC) of azobenzenes was determined for different compositions, indicating that small amounts of azobenzenes could induce gel formation, particularly in 5 wt% SA. Upon exposure to 365 nm UV light, the SMHGs exhibited reversible gel–sol transitions, underscoring their photoresponsive nature. This research offers valuable insights into the synthesis and photoresponsive properties of cationic, water-soluble azobenzenes, as well as their potential application in the development of photoresponsive hydrogels.

## 1. Introduction

Hydrogels are generally hydrophilic systems characterized by highly crosslinked structures associated with ions or crosslinking molecules. In the last decade, their use has increasingly extended into biomedical systems, finding applications in tissue engineering [1], biosensors, and drug delivery vehicles [2].

A hydrogel-based biomaterial may successfully release active compounds such as antibiotics and drugs encapsulated within it. Due to its high water content, it is like living tissue, has good malleability, and is non-adhesive. Hydrogels can also donate water to a wound site and maintain a wet environment for cell migration, thus promoting faster wound healing [3]. 

The development of so-called *smart* biomaterials is increasingly the focus of research into new systems that can be variably functionalized to respond to external stimuli (pH, temperature, light, redox) [4,5,6,7]. Light is the most widely used stimulus: safe and non-invasive, it can be easily introduced on demand, and the light source can have a specific wavelength and intensity that can be modulated [8]. In recent years, the development of smart biomaterials has gained significant attention due to their ability to respond dynamically to external stimuli such as temperature, pH, and light [7]. These materials are increasingly recognized for their potential in creating highly efficient, targeted therapeutic delivery systems and regenerative medicine applications.

Light-responsive biomaterials are commonly prepared by incorporating photoreactive molecules, such as photochromic compounds, into polymer matrices [9,10,11,12].

Within the field of photoresponsive molecules, azobenzene has been the most widely used photosensitive component for the design of photoresponsive biomaterials due to an almost degradation-free but mostly reversible isomerization process between the *cis* and *trans* isomers. Photoisomerization of azobenzene can be achieved by selective excitation of the *trans* isomer in the UV range (approx. 365 nm) and the *cis* isomer in the visible range (450 nm) [13,14]. When this molecule is introduced into polymer matrices, the photoisomerization of azobenzene can lead to structural modifications of the material, such as the gel–sol transition of hydrogels [15,16,17,18]. Therefore, these materials have been widely used as actuators for the design of optical and electro-optical devices in the biomedical field, in photocontrolled drug delivery, or in tissue engineering [15,19,20,21,22,23,24]. 

Recently, more attention has been given to supramolecular hydrogels (SMHGs) with low molecular weight (LMW) azobenzene as crosslinkers to make photoreactive SMHGs [25,26,27,28]. Supramolecular systems offer significant advantages over covalently crosslinked systems because they are more dynamic and formed by interchangeable interactions, such as hydrogen bonds, π-π stacking, and ionic interactions [29,30,31]. While the use of azobenzenes as linkers is commonly reported in the literature for organogel construction [32,33,34], cases of water-based gels containing azobenzenes are rarer due to their low hydrophilicity and adverse synthetic procedure. To facilitate their solubility in water, in some cases, azobenzenes have been modified with peptides [35,36,37], sugars [38,39], or even azobenzene-based surfactants [17,40,41]. In addition, most of these systems are based on chemical modification of polymers like polyacrylates, polyamides, or alginate with cyclodextrins that, binding with azobenzene, can cause macroscopic changes of polymers’ physical properties [15,42,43,44,45].

This work reports the design and synthesis of water-soluble azobenzenes AZO A, AZO B, and AZO C with different ionic head groups. The structural features and photoisomerization behaviors of these three azobenzenes were studied in detail by UV–Vis and ^1^H NMR spectroscopy. Furthermore, these azobenzenes were successfully used as crosslinkers to build photoresponsive SMHGs by electrostatic interactions with sodium alginate (SA) and π-π stacking interactions between aromatic rings of azobenzene moieties (Figure 1) that stabilized the crosslinked structures of the gels that are more versatile and easily tunable than hydrogels based on covalent bonds [46]. The hydrogels were characterized by FT-IR spectroscopy and SEM.

In recent advancements within alginate-based materials, complex synthesis processes have often posed significant barriers to practical application and scalability. For instance, the work [27], which involves an intricate eight-step synthesis incorporating expensive protecting groups, highlights the need for simpler, more cost-effective methods. Our research aims to streamline the fabrication process by employing easy-to-synthesize crosslinking monomers. This approach reduces production costs and enhances the feasibility of using these materials in real-world applications. By simplifying the synthesis pathway, we aim to make the advantages of innovative alginate-based materials more accessible and practical for broader applications.

## 2. Materials and Methods

All the reagents and solvents were purchased from Sigma Aldrich (Milan, Italy) and used without further purification. 

UV–Vis absorption spectra of the samples were recorded at 25 °C at 10^−5^ M concentration in Milli-Q water on an Evolution 201 spectrophotometer (Thermo Scientific, Rodgau, Germany). The spectral region 600–300 nm was investigated by using a cell path length of 1.0 cm. A UV–Vis lamp with a fixed wavelength was used to investigate the isomerization process of azobenzenes in solution and the gel–sol transitions of the hydrogels. The wavelength used for the measurements was 365 ± 2 nm with a power of 50 W.

The ^1^H NMR spectra were recorded with a DRX 400 spectrometer (Bruker, Billerica, MA, USA). The spectra were referenced to residual CHCl_3_ (^1^H: δ = 7.26 ppm), CH_3_OH (^1^H: δ = 3.33 ppm, 4.87 ppm), (CH_3_)_2_SO (^1^H: δ = 2.50 ppm), or H_2_O (^1^H: δ = 4.80 ppm) as indicated. The following abbreviations are used to express spin multiplicities in ^1^H NMR spectra: s = singlet; d = doublet; dd = double doublet; t = triplet; m = multiplet. For the isomerization study by ^1^H NMR, the sample was dissolved in deuterium oxide, first kept in the dark, and then irradiated with the UV–Vis lamp for 1 h and 3 h.

High-resolution mass spectra were acquired on an LTQ-Orbitrap instrument (Thermo-Fisher, Waltham, MA, USA) operating in positive ion mode. The molecules were dissolved in methanol or water at a concentration of 0.1 mg/mL and injected into the MS ion source. Spectra were acquired in the 150–900 *m*/*z* range. 

The infrared spectra were acquired by an FT-IR spectrometer (Spectrum Two FT-IR, Perkin Elmer, Waltham, MA, USA) at a resolution of 2.0 cm^−1^, 128 scans, and within a wavelength range of 4000–600 cm^−1^. The samples were analyzed in powder form; the hydrogels were pre-dried.

SEM analyses were performed using a scanning electron microscope, the Leo 1530 Gemini by Zeiss (Carl Zeiss NTS GmbH, Oberkochen, Germany), on the spun films; the operating voltage was 6 kV for all the measurements performed. The hydrogels, composed of 5 wt% SA and 1 wt% AZO, were dried before measurement. 

### 2.1. Synthesis of AZO A

All the synthesis patterns and identification numbers of the synthesized molecules can be found in the Appendix A.

#### 2.1.1. Synthesis of 4,4′-(Diazene-1,2-diyl)diphenol (**1** in Appendix A)

First, 1.00 g of 4-aminophenol (0.009 mol) was dissolved in an acid solution containing 6.6 mL of distilled water and 1.4 mL of 37% HCl. The suspension was stirred until the reagent was completely solubilized, for approximately 20 min, while maintaining a constant temperature in an ice-water bath (0–5 °C). Next, 0.695 g NaNO_2_ (0.010 mol) was dissolved in 1.1 mL of distilled water. The resulting solution was added slowly, drop by drop, to the acid solution containing the aminophenol (Solution A). Meanwhile, a basic solution at pH = 14 was prepared by dissolving 0.120 g NaOH in 8.62 mL of distilled water. The resulting basic solution was used to solubilize phenol (0.862 g; 0.009 mol) (Solution B). Solution A was added drop by drop to Solution B over a period of half an hour. The reaction mixture was left to stir for about three hours, keeping the temperature (12–15 °C) and pH (11) constant. The solution was then neutralized with acetic acid to pH = 5–6, resulting in the formation of a precipitate. The precipitate was vacuum filtered, washed with water, and crystallized in EtOH/H_2_O. The product obtained was a dark, fine, powdery solid. Yield: 30%. ^1^H NMR (400 MHz, MeOD): δ 6.97–6.99 ppm (d, 4H), 7.82–7.84 ppm (d, 4H).

#### 2.1.2. Synthesis of 1,2-Bis(4-((6-bromohexyl)oxy)phenyl)diazene (**2** in Appendix A)

First, 0.256 g of compound **1** (0.00119 mol), 0.657 g of K_2_CO_3_ (0.00476 mol), and 1.1 mL of 1,6-dibromohexane (0.00714 mol) were added to 12 mL of acetone in a double-necked flask and left for 24 h at reflux (60 °C) under nitrogen atmosphere. The resulting solution was cooled to room temperature, and the solvent was removed by a Rotavapor. The resulting solid was washed with hexane, followed by water to remove excess alkylating agents and salts, and then vacuum filtered. The product obtained was a dark orange solid. Yield: 98%. ^1^H NMR (400 MHz, CDCl_3_): δ 1.44–1.47 ppm (m, 8 H), 1.74–1.80 ppm (m, 4H), 1.83–1.88 ppm (m, 4H), 3.35–3.38 ppm (t, 4H), 3.96–3.99 ppm (t, 4H), 6.90–6.93 ppm (d, 4H), 7.78–7.80 ppm (d, 4H).

#### 2.1.3. Synthesis of 6,6′-((Diazene-1,2-diylbis(4,1-phenylene))bis(oxy))bis(N,N,N-trimethylhexan-1-aminium) (AZO A, Appendix A)

First, 0.300 g of compound **2** was dissolved in trimethylamine solution (18 mL) in a pressure tube. The solution was stirred for 48 h at 50 °C. The resulting mixture was poured into 250 mL diethyl ether, and the precipitate obtained was vacuum filtered and washed several times with diethyl ether to obtain a soluble dark yellow solid. Yield: 97%. ^1^H NMR (400 MHz, D_2_O): δ 1.34–1.41 ppm (m, 4H), 1.44–1.52 ppm (m, 4H), 1.71–1.81 ppm (m, 8H), 3.03 ppm (s, 18H), 3.24–3.25 ppm (t, 4H), 4.09–4.13 ppm (t, 4H), 7.08–7.10 ppm (d, 4H), 7.76–7.78 ppm (d, 4H). HRMS (ESI+) for C_30_H_50_N_4_O_2_^2+^: m/2z = 249.19; found: 249.19. ^1^H NMR of AZO A is reported in the Appendix A.

### 2.2. Synthesis of AZO B

The same procedure was followed for the synthesis of compounds **1** and **2** (Appendix A) as for synthesizing AZO A. 

#### Synthesis of 1,1′-(((Diazene-1,2-diylbis(4,1-phenylene))bis(oxy))bis(hexane 6,1diyl))bis(pyridin-1-ium) (AZO B, in Appendix A)

First, 50 mg of compound **2** (0.105 mmol) was dissolved in 3 mL of anhydrous CH_3_CN, and then 50 μL of pyridine (0.62 mmol) was added. The solution was stirred at 70 °C for 48 h. After the reaction, the solvent was removed by evaporation under reduced pressure. The crude product was washed with ethyl acetate, and the product was obtained as a brown powder with a yield of 68%. ^1^H NMR (400 MHz, D_2_O): δ 1.31–1.36 ppm (m, 4H), 1.41–1.49 ppm (m, 4H), 1.70–1.76 ppm (m, 4H), 1.94–2.01 ppm (m, 4H), 4.04–4.08 ppm (t, 4H), 4.53–4.56 ppm (t, 4H), 7.03–7.05 ppm (d, 4H), 7.74–7.76 ppm (d, 4H), 7.94–7.97 ppm (t, 4H), 8.38–8.46 ppm (m, 2H), 8.74–8.76 ppm (d, 4H). HRMS (ESI+) for C_34_H_42_N_4_O_2_^2+^ m/2z = 269.16; found: 269.16. ^1^H NMR of AZO B is reported in the Appendix A.

### 2.3. Synthesis of AZO C

#### 2.3.1. Synthesis of N-(4-((4-hydroxyphenyl)diazenyl)phenyl)acetamide (**1**′ in Appendix A)

First, 0.400 g of 4-aminoacetalinide (0.0027 mol) was dissolved in an acid solution containing 5.4 mL of distilled water and 570 μL of 37% HCl. The suspension was stirred until the reagent was completely solubilized, for approximately 20 min, while maintaining a constant temperature in an ice-water bath (0–5 °C). Next, 200 mg NaNO_2_ was dissolved in 1.2 mL of distilled water. The resulting solution was added slowly, drop by drop, to the acid solution containing the diazonium salt (Solution A). Meanwhile, a basic solution at pH = 14 was prepared by dissolving 0.103 g NaOH in 15 mL of distilled water. The resulting basic solution was used to solubilize phenol (0.254 g; 0.003 mol) (Solution B). Solution A was added drop by drop to Solution B over half an hour. The reaction mixture was left to stir for about three hours, keeping the temperature (12–15 °C) and pH (11) constant. The solution was then neutralized with acetic acid to pH = 5–6, resulting in the formation of a precipitate. The precipitate was vacuum filtered and washed with water. The product obtained was a dark, fine, powdery solid. Yield: 77%. ^1^H NMR (400 MHz, MeOD): δ 2.18 ppm (d, 3H), 6.91–6.93 ppm (d, 2H), 7.72–7.73 ppm (d, 2H), 7.80–7.85 ppm (q, 4H).

#### 2.3.2. Synthesis of N-(4-((4-(4-bromobutoxy)phenyl)diazenyl)phenyl)acetamide (**2**′ in Appendix A)

Compound **1**′ (250 mg, 0.98 mmol), 1,4-dibromobutane (1.17 mL, 9.8 mmol), KI (16.3 mg, 0.098 mmol), and K_2_CO_3_ (271 mg, 1.96 mmol) were dissolved in 10 mL of acetone and the reaction mixture was left to reflux for approximately 4 h. After this time, the mixture was cooled, and an extraction was performed to recover the product using a solution of brine and ethyl acetate. The organic phase was recovered and dried and the solvent evaporated under vacuum. The crude solid was purified by a chromatographic column using hexane: ethyl acetate (60:40). The product was a yellow-orange solid, with a yield of 87%. ^1^H NMR (400 MHz, MeOD): δ 1.87–1.91 ppm (m, 2H), 1.96–2.01 ppm (m, 2H), 2.06 ppm (s, 3H) 3.43–3.47 ppm (t, 3H), 4.01–4.04 ppm (t, 3H), 6.95–6.98 ppm (d, 2H), 7.62–7.64 ppm (d, 2H), 7.73–7.78 ppm (q, 4H).

#### 2.3.3. Synthesis of 4-(4-((4-Acetamidophenyl)diazenyl)phenoxy)-N,N,N-trimethylbutan-1-aminium (**3**′ in Appendix A)

Compound **2**′ (200 mg) was dissolved in trimethylamine solution (12 mL) in a pressure tube. The solution was stirred for 48 h at 50 °C. The resulting mixture was added to diethyl ether (250 mL), and the precipitate obtained was vacuum filtered and washed several times with diethyl ether to obtain a yellow solid. Yield: 92%. ^1^H NMR (400 MHz, DMSO): δ 1.78–1.82 ppm (m, 2H), 1.85–1.91 ppm (m, 2H), 2.10 (s, 3H) 3.07 ppm (s, 9H), 3.36–3.40 ppm (t, 2H), 4.13–4.16 ppm (t, 2H), 7.13–7.15 ppm (d, 2H), 7.78–7.83 ppm (q, 4H), 7.86–7.88 ppm (d, 2H), 10.27 ppm (s, 1H).

#### 2.3.4. Synthesis of 4-(4-((4-Aminophenyl)diazenyl)phenoxy)-N,N,N-trimethylbutan-1-aminium (AZO C)

Compound **3**′ (175 mg) was dissolved in 37% HCl (13 mL) and stirred overnight at 40 °C. The acid was then removed by several additions of distilled water and subsequent evaporation of the solvent, resulting in a dark solid. Yield: 50%. ^1^H NMR (400 MHz, DMSO): δ 1.75–1.81 ppm (m, 2H), 1.86–1.92 ppm (m, 2H), 3.08 (s, 9H) 4.10–4.13 ppm (t, 2H), 7.02–7.04 ppm (d, 2H), 7.07–7.09 ppm (d, 2H), 7.62–7.65 ppm (d, 2H), 7.75–7.77 ppm (d, 2H). HRMS (ESI+) for C_19_H_27_N_4_O^+^ *m*/*z*: 327.22 found: 327.25. ^1^H NMR of AZO C is reported in the Appendix A.

### 2.4. Hydrogel Preparation

A specific amount of sodium alginate (SA) (20, 50, or 80 mg) was dissolved in 1 mL of Milli-Q water or 1 mL of acetic acid solution of pH 4 and heated to 60 °C until the polymer was completely dissolved. Next, azobenzene (5 or 10 mg) was added, and the solution was allowed to cool to room temperature. The hydrogels formed upon cooling the solutions.

## 3. Results and Discussion

### 3.1. Synthesis of Ionic Azobenzenes

The chemical structures of AZO A, AZO B, and AZO C are depicted in Figure 2.

Compounds **1** and **1**′ (Appendix A) were obtained by diazo-coupling reaction [47], forming the diazonium salt in acid media in the presence of sodium nitrite, followed by subsequent reaction with phenol in alkaline media. For AZOs A and B, compound **1**, which has two hydroxyl groups in 4,4′ positions on the aromatic rings, is reacted with the alkylating agent 1,6-dibromohexane, and the final products, azobenzenes A and B, gain two charged heads on opposite sides of the azobenzene core [48] by reaction with different amines: AZO A has trimethylammonium heads, AZO B pyridinium heads. In AZO C, compound **1**′ has a hydroxyl group and an amide group in the para position, obtained from phenol and amino acetanilide. Compound **2**′ (Appendix A) was obtained by the reaction with 1,4-dibromobutane and the introduction of a cation head with trimethylamine. Finally, the acid reaction reduced the amide group to amine [49]. 

The products of all synthetic steps were purified and characterized. ^1^H NMR spectroscopy and mass spectrometry confirmed the structure of three cationic azobenzenes.

### 3.2. Light-Responsive Behavior of Azobenzenes

The isomerization study recorded UV–Vis absorption and ^1^H NMR spectra of the synthesized azobenzenes.

For UV–Vis analysis, the azobenzenes were dissolved in water at a concentration of 10^−5^ M. Absorption spectra were recorded after keeping the samples in the dark for about 2 h to promote complete conversion to the *trans* isomer. Then, the solutions were irradiated by UV light at 365 nm to induce *trans–cis* isomerization. Absorption spectra were recorded at different irradiation times. Relaxation times were measured for the samples in the dark to assess the time required for the transition to the *trans* form. For AZO A, complete *trans–cis* isomerization was recorded after 30 s of irradiation with UV light at 365 nm (Figure 3a). The spectra show a decrease and shift of the band at about 365 nm related to the π→π* transition with an increase in the band at 450 nm, relative to the n→π* transition. A complete return to the *trans* form was achieved in 20 h in the dark (Figure 3b).

Also, for AZO B, *trans–cis* isomerization under UV irradiation at 365 nm was very fast, completed in 1 min (Figure 4a), while the transition of the *cis* to the *trans* form occurred in 15 h in the dark (Figure 4b).

The absorption spectrum of AZO C (Figure 5a) showed a shift of about 25–30 nm to longer wavelengths for the band related to the π→π* transition, leading to overlap between the two transitions due to the presence of the strong electron-donating amine group in the para position [14,50].

The presence of the amine group, similarly, also affected isomerization and relaxation times: the *trans* to *cis* isomerization process for AZO C (Figure 5a) was performed with irradiation at 365 nm in 10 min. The return to the *trans* form (Figure 5b) occurred in about 1 h, faster than for the previous azobenzenes, probably due to the higher stability of the *trans* isomer.

Due to the long relaxation times of AZO A and AZO B, we also studied the photoisomerization process by ^1^H NMR spectroscopy. As an example, we have reported the ^1^H NMR isomerization study of AZO A. The sample, solubilized in D_2_O, was placed in the dark for 2 h to ensure complete conversion of the molecule to the *trans* isomer (Figure 6, blue line). A first ^1^H NMR spectrum of the sample irradiated with 365 nm UV light for about 1 h (green line), followed by a second spectrum after 3 h of irradiation (red line), was recorded.

As can be seen in Figure 6, the two isomers exhibit a different proton spectrum, particularly in the aromatic zone (6.5–8.0 ppm) (magnification on the left). The spectrum recorded in the dark shows only the peaks for the *trans* isomer, **1*t***, **2*t***, and **3*t***. After 1 h of UV light irradiation (green line), a new set of proton signals, **1–3*c***, suggested a *trans* to *cis* isomerization. By integrating the proton signals of **2*t***/**2*c*** or **3*t***/**3*c***, the *trans:cis* ratio of the two isomers could be estimated as 49:51. After 3 h of UV light irradiation at 365nm (red line), the peaks for the *cis* isomer, **1–3*c*** increase, while those for the *trans* isomer decrease, indicating a higher shift in equilibrium towards the *cis* isomer. Integrating the proton signals, the *trans:cis* ratio of the two isomers could be estimated as 15:85. 

### 3.3. Hydrogel Formation and pH-Responsive Properties

Alginate is one of the most used natural polymers to form hydrogels characterized by reversible gelation in aqueous solution in the presence of bivalent cations, such as calcium, barium, and magnesium, via ionic crosslinking with carboxyl acid groups of alginate [51]. In this work, the synthesized azo molecules have been used as potential crosslinkers for developing photoresponsive supramolecular hydrogels (SMHGs) by electrostatic-type interactions with sodium alginate (SA). We first studied the minimum gelation concentration (MGC), which is the minimum concentration of gelator (azobenzene) required to form a stable gel [17], in two different pH conditions: pH = 7 and pH = 4. 

The SA-AZO hydrogels are stable at cold and room temperatures up to the upper critical solution temperature (UCST) phase transition. On heating to around 55 °C, the polymer networks slowly become soluble due to decreased hydrophobic interactions. At temperatures lower than the UCST, the electrostatic interactions between the di-cationic azo crosslinkers and the alginate chains are stable. When the temperature rises above the UCST, these interactions decrease in favor of interactions with water [7].

The different concentrations of the two components of the hydrogels are shown in Table 1 at pH = 7 and Table 2 at pH = 4. At pH = 4, the amine group in AZO C is protonated, while the carboxyl groups of the alginate remain deprotonated. The pK_a_ value of alginate carboxyl groups is 3.5 [51,52]. Sodium alginate (SA) was dissolved in water or an acidic solution at pH = 4 at 60 °C. Azobenzene was added while the solution was stirred at 60 °C. Once the solubilization of azo was complete, the mixture was allowed to cool to room temperature until hydrogel formation (approximately a couple of hours), confirmed by the vial inversion method. Table 1 and Table 2 present the hydrogels’ physical states and compositions under two different pH conditions.

As shown in Table 2, the three azobenzenes form stable hydrogels with sodium alginate (SA) under acidic conditions at pH 4. With AZO A, stable hydrogels are formed using 1 wt% (20 mM) of azobenzene with 5 wt% of SA (Figure 7a, vial 3). Increasing the amount of SA to 8 wt%, 0.5 wt% of AZO A (10 mM) is enough to obtain a gel (Figure 7a, vial 2). For AZO B, 0.5 wt% (9 mM) with 5 wt% SA is sufficient for gel formation (Figure 7b, vial 5), probably due to additional π-π interactions between the pyridinium rings. Also, for AZO C, hydrogels were obtained using 0.5 wt% (15 mM) or 1 wt% (30 mM) of azobenzene with 5 wt% of SA (Figure 7c, vials 9 and 10).

The gels show a pH-dependent behavior typical of ionic polymers, leading to swelling and collapse under pH variations [53]. As the pH changes, the charged groups of the polymer chains and the azobenzene crosslinkers may undergo protonation or deprotonation, leading to a change in the electrostatic interactions and causing the gel structure to collapse [52].

Gel formation occurs when the alginate chains are close enough together. Two essential aspects are the number of negative charges on the polymer chain and the counterions. At very acidic pHs, there are no carboxylate groups; therefore, gel formation is impossible, whereas, at basic pHs, there are too many negative charges. Mono-cations effectively approach the chain’s carboxylate groups but cannot coordinate multiple groups, so mono-cations do not induce gel formation. Di- or tri-cations are effective as long as their concentration does not become too high. AZO A and AZO B molecules are large di-cations. They can coordinate charged polymer chains, but if their concentration is lower than that of the carboxylate groups, neutralization is achieved with the sodium ions present, which are unsuitable for gelation. For this reason, gel formation occurs only in a narrow pH range. Therefore, gels with AZO A and AZO B remain stable between pH values of 4 and 7.

Regarding the gels with AZO C, the terminal amine group of AZO C is protonated at pH values below 5, so the corresponding alginate gel is stable in pH ranges between 5 and 3, when the AZO C crosslinker is a di-cation and can effectively crosslink the alginate polymer chains.

This pH-dependent behavior makes the hydrogels potentially valuable materials for drug delivery to different human body regions.

### 3.4. Hydrogel Photoresponsive Properties

The gels were irradiated with UV light at 365 nm to study photoresponsivity. After a few minutes of irradiation, the hydrogels with AZO A and AZO B showed a transition from gel to sol, following *trans*-to-*cis* isomerization of the azobenzene. For hydrogels with AZO C, an irradiation time of about 30 min at 365 nm was required for the gel–sol transition (Figure 8).

The *trans* to *cis* azobenzene isomerization causes the SMHG network to collapse due to the disruption of electrostatic interactions between azobenzenes and SA, transforming SMHGs from gels to solutions. All hydrogels return to the gel state in a few minutes under daylight at room temperature due to the return of azobenzenes to the more stable *trans* form. The behavior of the hydrogels under the influence of pH and light is summarized in the Appendix A. 

The reversible and controllable light-responsive behavior demonstrated by the azobenzene-incorporating hydrogels signifies their potential as smart biomaterials. As highlighted by [7], such materials are crucial for the next generation of medical treatments, offering precise control over drug release rates and enhanced adaptability in tissue engineering contexts.

### 3.5. FT-IR Characterization of the Hydrogels

The study of the chemical structure of alginate–azobenzene hydrogels is critical to investigating the plausible chemical interactions among the compounds. Figure 9a–c show the IR spectra of SA and AZOs A, B, and C as the pure components and the corresponding hydrogels. Pure sodium alginate (SA), red curves in spectra a, b, and c of Figure 9, presents characteristic absorption bands: around 1600 cm^−1^ and 1412 cm^−1^, corresponding to the asymmetric and symmetric stretching vibration of the -COO^−^ groups, respectively; the single-bond -CH- vibration band occurs at 2936 cm^−1^ and 1294 cm^−1^. The intense band from the -C-O-C- stretching occurs around 1024 cm^−1,^ and a large absorption band around 3200 cm^−1^ is due to the stretching vibration of the OH group [54]. The AZO A spectrum (black curve, Figure 9a) is characterized by symmetric and asymmetric stretching vibration of sp^2^ and sp^3^ -CH at 3450–3380 cm^−1^ and 2940–2865 cm^−1^, a series of peaks at 1598–1578 cm^−1^ due to stretching vibration of aromatic rings, and the stretching vibration of the -N=N- double bond at 1492 cm^−1^, which is partially overlapped with deformation bands of methylene groups attached to the cationic N^+^ at 1475 cm^−1^ [55]. Two strong peaks at 1238 and 1144 cm^−1^ are due to the stretching of -C-N- aromatic and aliphatic bonds. The AZO B spectrum (grey curve in Figure 9b) also shows symmetric and asymmetric stretching vibration of sp^2^ and sp^3^ -CH in the range between 2800 and 3000 cm^−1^. In the region of 1600–1400 cm^−1^ there are a series of peaks due to the stretching vibration of aromatic rings of azobenzene nuclei and of pyridinium rings, and at 1242 and 1147 cm^−1^, the stretching of -C-N- aromatic and aliphatic bonds. The AZO C spectrum (yellow curve in Figure 9c) is characterized by symmetric and asymmetric stretching vibration of -CH and -NH_2_ groups in the region of 3400–2800 cm^−1^, the latter accompanied by bending vibration peaks at 1650 cm^−1^ and 830 cm^−1^, confirming the presence of a primary amine. Again, there are -N=N- double-bond stretching bands at 1547 cm^−1^, a series of peaks at 1504–1480 cm^−1^ due to benzene ring stretching, and -C-N- (aliphatic and aromatic) bonds at 1379 cm^−1^ and 1253 cm^−1^. In the spectrum of the SA-AZO A hydrogel (blue curve, Figure 9a), the broadening of the band relative to the OH groups of the alginate is evidence of the formation of hydrogen bonds between polymer chains. The bands corresponding to the -COO^−^ (1600 cm^−1^ and 1412 cm^−1^) of SA tend to be less intense, and there is a slight decrease in the wavenumber due to ionic interaction with azobenzene, as reported in the literature for the electrostatic interactions of -COO^−^ groups of alginate with Ca^2+^ ions [54] and in alginate/chitosan crosslinked hydrogels [56,57]. The same behavior can be observed for the band related to the stretching vibration of the -C-O-C- bonds. The weak peak at 1246 cm^−1^ can be attributed to the stretching vibration of the -CH bond of SA or the band of the -C-N- bond of azobenzene. In both cases, a decrease in intensity and a shift to a lower wavenumber indicate the presence of interactions between the two components.

In the hydrogel formed by SA with AZO B (purple curve in Figure 9b), as for the hydrogel SA-AZO A, we can see the same shift in the wavenumbers and decrease in intensities of the bands. In the hydrogel formed by SA and AZO C, the azobenzene amine is protonated, and this is confirmed by the presence in the IR spectrum (green curve, Figure 9c) of a band formed by two peaks relating to the vibration stretching of the -N-H bond of the -NH_3_^+^ group at 3300–3190 cm^−1^, accompanied by bands at 1620 cm^−1^ overlapped with the band relating to the carboxylic group of the alginate and 1400 cm^−1^. The IR absorption peaks of the main functional groups of the corresponding azobenzenes and hydrogels are summarized in the Appendix A.

### 3.6. Hydrogel Morphology

The morphology of the hydrogels was investigated by SEM. The samples analyzed were composed of 5 wt% SA and 1 wt% AZO.

Figure 10 shows the SEM micrographs of the hydrogel surfaces acquired at different magnifications: 20,000× and 50,000×. The surface of the hydrogels is quite rough, with uneven micropore distribution with irregular shape and size. Specifically, SEM surface images at 50,000× show a higher structuring for the gel with AZO A, which is characterized by clear patterning with rather neat cavities. The structuring is less evident in the case of the sample with AZO B, which shows a rough surface but with fewer evident cavities. Finally, in the case of the gel with AZO C, a flat surface is observed, and no organized cavities are evident. This difference can be ascribed to the different lengths of the three azo-based linkers, two of which (AZOs A and B) possess two long tails with terminal charges and one (AZO C) which instead has only one charged tail and an amino group that is not separated from the aromatic system (see Figure 2). 

The porous structure offers an advantage for the use of obtained hydrogels as phototriggered vehicles for the delivery of drugs or biologically active compounds [42,58,59].

## 4. Conclusions

In this study, we have successfully synthesized and characterized photoresponsive supramolecular hydrogels (SMHGs) employing ionic azobenzenes, showcasing their significant potential for applications in smart materials. Incorporating azobenzenes, specifically AZO A, AZO B, and AZO C, into alginate matrices marks a novel approach to creating light-responsive hydrogels. Our findings highlight the control over the hydrogel’s synthesis, its reversible and controllable light-responsive behavior, and its morphological characteristics, as underscored by comprehensive FT-IR spectroscopy analysis. The tailored photoresponsivity of these compounds under different light stimuli and environmental conditions exemplifies their versatility in modulating the hydrogels’ properties, making them promising candidates for developing advanced polymers. These materials exhibit potential applications ranging from drug delivery and tissue engineering to environmental sensing, emphasizing their role in introducing innovative therapeutic delivery methods that respond to external light stimuli. 

The development of our light-responsive hydrogels underscores the substantial potential of integrating smart material characteristics into biomaterials. Future research will focus on tailoring these properties to specific clinical applications, such as controlled drug delivery systems and responsive tissue scaffolds, to fully exploit their smart capabilities. 

## Figures and Tables

**Figure 1 polymers-16-01233-f001:**
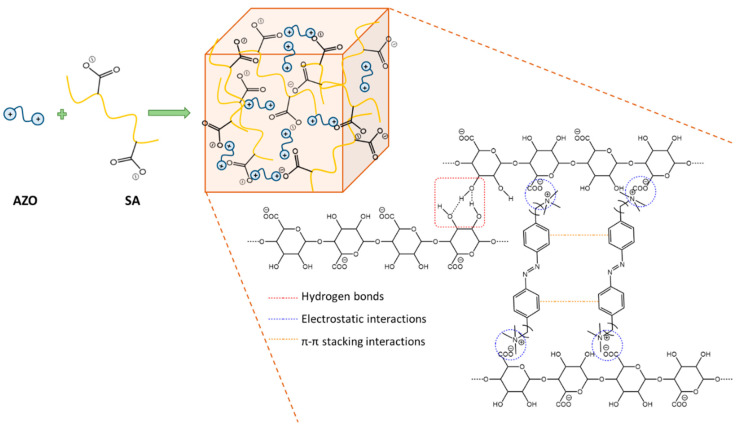
Schematic representation of alginate–azobenzene (AZO A)-based supramolecular hydrogels.

**Figure 2 polymers-16-01233-f002:**
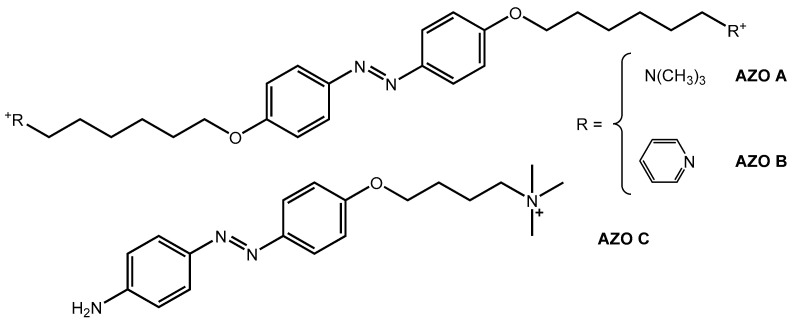
Chemical structure of AZO A, AZO B, and AZO C.

**Figure 3 polymers-16-01233-f003:**
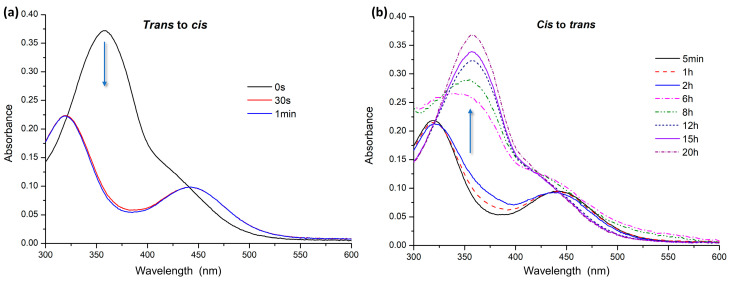
UV–Vis spectra of AZO A: (**a**) *trans* to *cis* isomerization under UV irradiation; (**b**) *cis* to *trans* relaxation in the dark.

**Figure 4 polymers-16-01233-f004:**
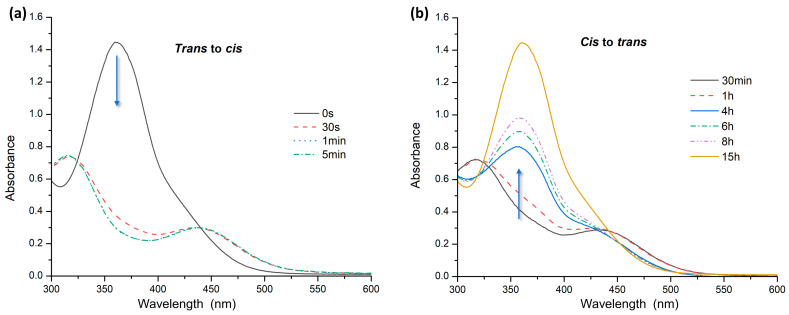
UV–Vis spectra of AZO B: (**a**) *trans* to *cis* isomerization under UV irradiation; (**b**) *cis* to *trans* relaxation in the dark.

**Figure 5 polymers-16-01233-f005:**
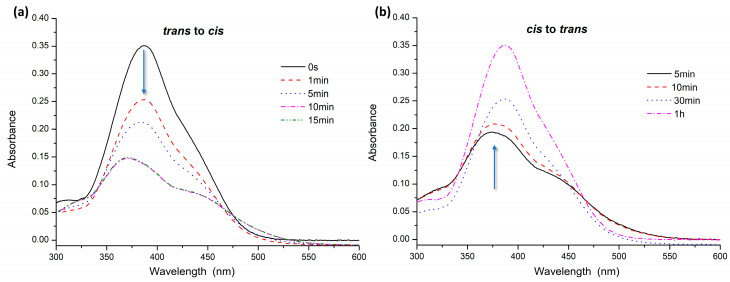
UV–Vis spectra of AZO C: (**a**) *trans* to *cis* isomerization under UV irradiation; (**b**) *cis* to *trans* relaxation in the dark.

**Figure 6 polymers-16-01233-f006:**
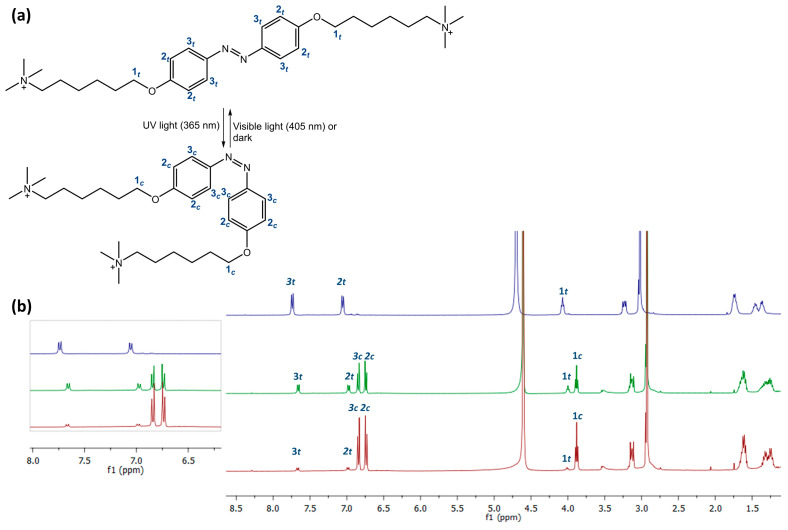
(**a**) Isomerization process of AZO A; (**b**) ^1^H NMR spectra of AZO A after different irradiation times. In the molecules and spectra, the protons of the two *cis* (***c***) and *trans* (*t*) isomers are indicated.

**Figure 7 polymers-16-01233-f007:**
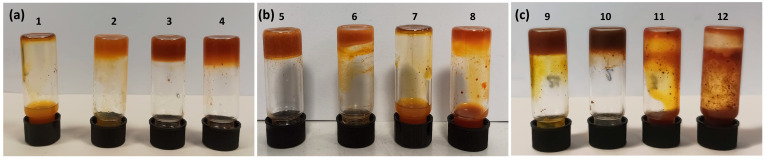
Pictures of hydrogels’ physical states at different concentrations of (**a**) AZO A; (**b**) AZO B; and (**c**) AZO C with SA. The numbers on the vials correspond to the concentrations given in Table 2.

**Figure 8 polymers-16-01233-f008:**
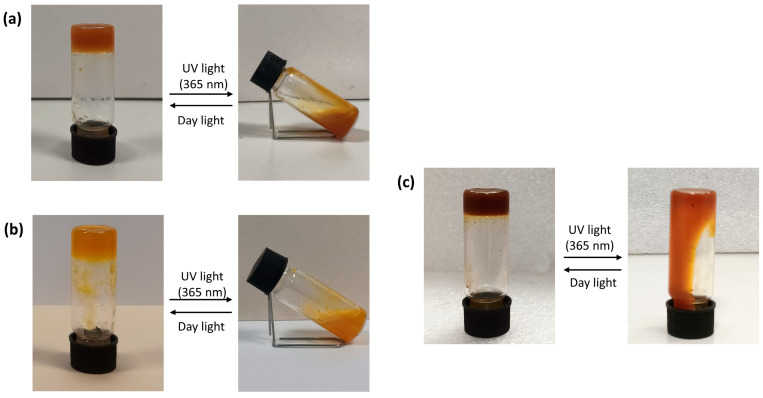
Pictures of hydrogels after UV light and daylight irradiation and gel–sol transition process: (**a**) 1wt% AZO A + 5wt% SA; (**b**) 0.5wt% AZO B + 5wt% SA; (**c**) 0.5wt% AZO C + 5wt% SA.

**Figure 9 polymers-16-01233-f009:**
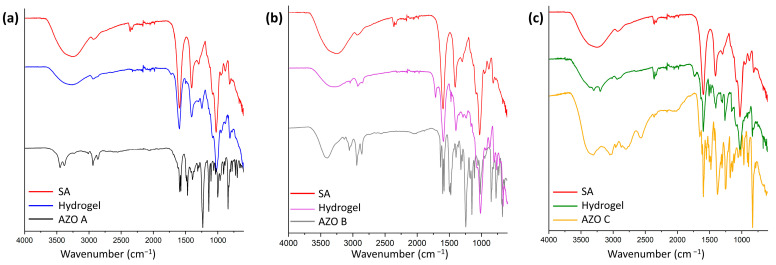
FT-IR spectra of the hydrogels (5 wt% SA and 1 wt% AZO) and pure components: (**a**) AZO A, (**b**) AZO B, and (**c**) AZO C.

**Figure 10 polymers-16-01233-f010:**
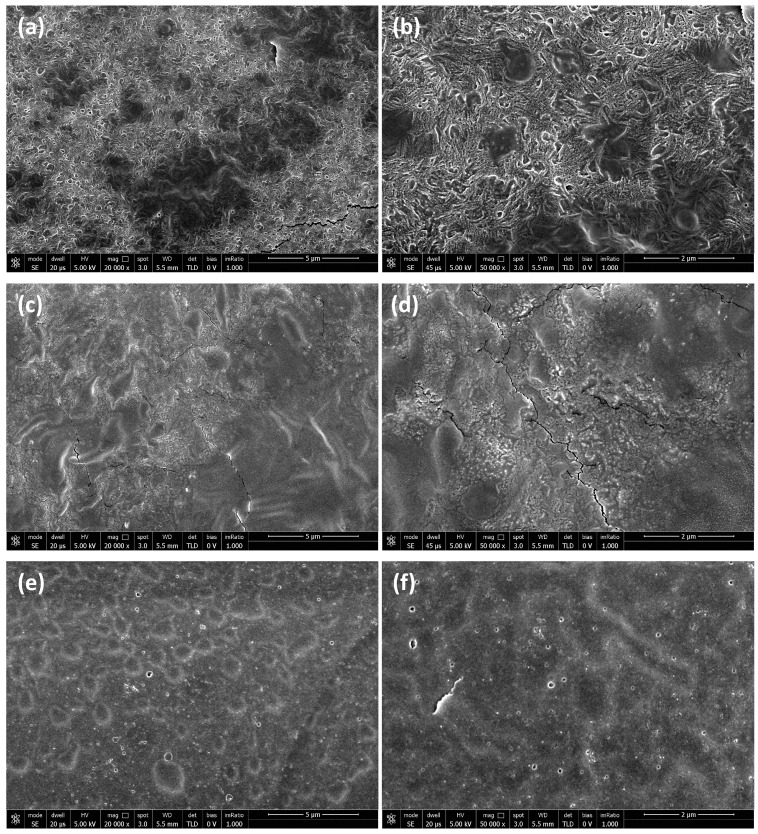
SEM micrographs of the hydrogel surfaces with (**a**,**b**) AZO A; (**c**,**d**) AZO B; (**e**,**f**) AZO C. The magnifications and scale bars are, respectively: (**a**,**c**,**e**) 20,000× and 5 μm; (**b**,**d**,**f**) 50,000× and 2 μm.

**Table 1 polymers-16-01233-t001:** Hydrogel weight % composition at pH 7.

% wt SA_% wt AZO	8%/0.5%	8%/1%	5%/0.5%	5%/1%
SA_AZO A	V	V	L	L
SA_AZO B	V	V	L	L
SA_AZO C	P	P	P	P

L: liquid; V: viscous; P: precipitate.

**Table 2 polymers-16-01233-t002:** Hydrogel weight % composition at pH 4.

% wt SA_% wt AZO	8%/0.5%	8%/1%	5%/0.5%	5%/1%	2%/0.5%	2%/1%
SA_AZO A	G (2)	G (4)	L (1)	G (3)	*	*
SA_AZO B	*	*	G (5)	G (6)	L (7)	V (8)
SA_AZO C	*	*	G (9)	G (10)	L (11)	L (12)

L: liquid; V: viscous; G: gel. * No additional tests were carried out beyond the MGC of crosslinking AZO. In brackets are the numbers of the vials, as shown in Figure 7.

## Data Availability

The data presented in this study are contained within the article and Appendix A.

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
