# Peer review of "Cationic Azobenzenes as Light-Responsive Crosslinkers for Alginate-Based Supramolecular Hydrogels"

_polymers, 2024, doi:10.3390/polym16091233_

Round 1
Reviewer 1 Report
Comments and Suggestions for Authors
The paper presents the fabrication of a new material: alginate-based supramolecular hydrogels with cationic azobenzenes as light-responsive crosslinkers. In addition to light-responsiveness, pH-responsiveness was also shown for this material. The paper is interesting, demonstrates novel findings, and can be accepted for publication after revision. The following issues should be clarified.
As the authors exactly noted, the developed system is "smart" and potentially can be applied as biomaterials, so this idea should be highlighted for understanding the potential of the applications. I suggest using some reviews, for example, https://doi.org/10.1002/tcr.202300217 to highlight the importance of this trend.
The methodology of hydrogel formation is absent in the methods.
It will be valuable to present the FTIR results in the form of a table with wavenumbers and assigned groups.
It will be valuable to depict the molecular behaviour of the hydrogels under the action of various pHs and lights in the form of a scheme.
The rheological analysis is recommended for a more comprehensive understanding of gel behaviour.
What will be the behavior of the hydrogel under real-world conditions (salts, etc.)? What about the temperature impact on the gel? At least an appropriate discussion should be introduced.
Comments on the Quality of English Language
Minor editing of English language required.
Author Response
REVIEWER 1
The paper presents the fabrication of a new material: alginate-based supramolecular hydrogels with cationic azobenzenes as light-responsive crosslinkers. In addition to light-responsiveness, pH-responsiveness was also shown for this material. The paper is interesting, demonstrates novel findings, and can be accepted for publication after revision. The following issues should be clarified.
As the authors exactly noted, the developed system is "smart" and potentially can be applied as biomaterials, so this idea should be highlighted for understanding the potential of the applications. I suggest using some reviews, for example, https://doi.org/10.1002/tcr.202300217 to highlight the importance of this trend.
We thank the Reviewer for the positive comments. We have expanded the Introduction and the Discussion sections, highlighting the potential application of these materials.
We have added the suggested review in the reference list.
The methodology of hydrogel formation is absent in the methods.
We have added the methodology of hydrogel formation in the Materials and Methods section.
It will be valuable to present the FTIR results in the form of a table with wavenumbers and assigned groups.
We have added a table in Supplementary Material with main functional groups and peaks of the three azobenzenes and their corresponding hydrogels.
It will be valuable to depict the molecular behaviour of the hydrogels under the action of various pHs and lights in the form of a scheme.
As suggested, we have added a Table in the Supplementary Material, with data about the molecular behavior of the hydrogels under the action of pHs and lights.
The rheological analysis is recommended for a more comprehensive understanding of gel behaviour.
We agree with the reviewer that rheological analysis is important to characterise the mechanical behaviour of the gels produced. We have performed some tests on our materials, but at the moment some gels are very soft, and it was not possible to perform a complete rheological analysis.
We aim to introduce new different types of crosslinkers into the gels in order to obtain stronger materials.
What will be the behavior of the hydrogel under real-world conditions (salts, etc.)? What about the temperature impact on the gel? At least an appropriate discussion should be introduced.
While the current study primarily focused on the photoresponsive behavior of the hydrogels, understanding their stability and performance in real-world conditions, such as varying ionic strengths and temperature ranges, is crucial. For instance, the presence of salts affects the electrostatic interactions within the hydrogel network, altering its structural integrity and responsiveness. For example Ca2+ ions can interact with alginate chains. In presence of calcium salts, this interaction can become predominant respect to azobenzene crosslinkers, leading to the irreversible formation of Ca/alginate hydrogels.
We performed stability experiments on the synthesized gels at different temperatures. The hydrogels behave as thermo-responsive polymers with UCST (Upper Critical Solution Temperature) phase transition.
We extended the discussion section 3.3 including these considerations.
Reviewer 2 Report
Comments and Suggestions for Authors
The research focus on stimuli-responsive hydrogels has been extremely popular recently. In particular, light-responsive hydrogels are of great interest due to their possible application in various science-intensive fields, especially in biomedical applications. The ammonium-containing derivatives investigated by the authors, along with light-responsive properties, may additionally possess antibacterial properties, which further stimulates the interest to this work. The reviewer believes that the work has no significant shortcomings, is very interesting and relevant and can be published after minor corrections. The English language does not raise any questions.
1. Supplementary materials contain no NMR spectra of intermediates and target azobenzenes.
2. It is not quite clear how the sentence on lines 229-230 about complete conversion to the trans form within 2 hours (before recording) and the data of Figures 3 and 4, according to which complete conversion occurs within 15-20 hours, fit together.
3. Line 305: the mention of Table 1 seems irrelevant in this context.
4. There is almost no discussion of the effect of pH on hydrogel formation (differences in the behavior of the SA-AZO systems at different pH).
Author Response
REVIEWER 2
The research focus on stimuli-responsive hydrogels has been extremely popular recently. In particular, light-responsive hydrogels are of great interest due to their possible application in various science-intensive fields, especially in biomedical applications. The ammonium-containing derivatives investigated by the authors, along with light-responsive properties, may additionally possess antibacterial properties, which further stimulates the interest to this work. The reviewer believes that the work has no significant shortcomings, is very interesting and relevant and can be published after minor corrections. The English language does not raise any questions.
We thank the Reviewer for the positive comments.
- Supplementary materials contain no NMR spectra of intermediates and target azobenzenes.
We have added the 1H NMR spectra of the azobenzenes AZO A, AZOB and AZO C in supplementary material. For all the intermediates the 1H NMR peak peaking is reported in the experimental section.
- It is not quite clear how the sentence on lines 229-230 about complete conversion to the trans form within 2 hours (before recording) and the data of Figures 3 and 4, according to which complete conversion occurs within 15-20 hours, fit together.
After dissolving the sample in water, the solution is first kept in the dark for two hours to promote complete conversion of the molecule to the trans form. The solution is then irradiated by UV light and the spectra of isomerisation to the cis form are recorded (Fig. 3 and 4 (a)). The times of 15 h and 20 h refer to the subsequent spontaneous return of the molecules to the trans form in the dark (Fig. 3 and 4 (b)).
We have rephrased the description in the text to clarify this procedure.
- Line 305: the mention of Table 1 seems irrelevant in this context.
Yes, we corrected the text accordingly.
- There is almost no discussion of the effect of pH on hydrogel formation (differences in the behavior of the SA-AZO systems at different pH).
We have added a new paragraph in the discussion section (3.3) where the behaviour of the three hydrogels at different pH was discussed.
Reviewer 3 Report
Comments and Suggestions for Authors
The authors reported the cationic azobenzene as light-responsive for Alginate-based supramolecular hydrogels. The manuscript was well written with all experiments carefully carried out. The authors showed a detailed characterization of the alginate-based materials with SEM, UV-vis measurement for the switching conformation between cis to trans and vice versa. The 1H NMR showed the chemical structure identification of carbon backbone of the AZO A chemical, and the FTIR confirmed the modification of sodium alginate and AZO within the hydrogels. The morphology of the hydrogel was also investigated by SEM.
However, the manuscript lacks innovation and application for the synthesized materials. Photo-responsive supramolecular hydrogels, especially alginate-based azobenzene gels, are well studied with many papers cover the synthesized, characterization of molecular structure, cis-trans conversion, and morphology. For examples: “Reversible Mechanical Regulation and Splicing Ability of Alginate-Based Gel Based on Photo-Responsiveness of Molecular-Level Conformation”, Materials (Basel). 2019 Sep; 12(18): 2919, doi: 10.3390/ma12182919; “Reversible photodissipation of composite photochromic azobenzene-alginate supramolecular hydrogels”, RSC Adv., 2022,12, 4771-4776, doi.org/10.1039/D1RA09218A; “Alginate-Based Smart Materials and Their Application: Recent Advances and Perspectives”, Topics in current chemistry, 2022, Volume 380, article number 3, Springer Link. Therefore, the manuscript should improve on the usage of the materials and re-submit with a better application for the materials.
Comments on the Quality of English LanguageThe manuscript was well written in English.
Author Response
REVIEWER 3
The authors reported the cationic azobenzene as light-responsive for Alginate-based supramolecular hydrogels. The manuscript was well written with all experiments carefully carried out. The authors showed a detailed characterization of the alginate-based materials with SEM, UV-vis measurement for the switching conformation between cis to trans and vice versa. The 1H NMR showed the chemical structure identification of carbon backbone of the AZO A chemical, and the FTIR confirmed the modification of sodium alginate and AZO within the hydrogels. The morphology of the hydrogel was also investigated by SEM.
However, the manuscript lacks innovation and application for the synthesized materials. Photo-responsive supramolecular hydrogels, especially alginate-based azobenzene gels, are well studied with many papers cover the synthesized, characterization of molecular structure, cis-trans conversion, and morphology. For examples: “Reversible Mechanical Regulation and Splicing Ability of Alginate-Based Gel Based on Photo-Responsiveness of Molecular-Level Conformation”, Materials (Basel). 2019 Sep; 12(18): 2919, doi: 10.3390/ma12182919; “Reversible photodissipation of composite photochromic azobenzene-alginate supramolecular hydrogels”, RSC Adv., 2022,12, 4771-4776, doi.org/10.1039/D1RA09218A; “Alginate-Based Smart Materials and Their Application: Recent Advances and Perspectives”, Topics in current chemistry, 2022, Volume 380, article number 3, Springer Link. Therefore, the manuscript should improve on the usage of the materials and re-submit with a better application for the materials.
We would like to thank the Reviewer for the comments and constructive criticism.
We would like to point out the innovative aspects of our work, compared to the references mentioned by the reviewer. Two of these references are already included in our manuscript precisely because they are relevant to our research.
In particular:
- In “Reversible Mechanical Regulation and Splicing Ability of Alginate-Based Gel Based on Photo-Responsiveness of Molecular-Level Conformation” by Xiaozhou Ma et al., the materials undergo a chemical crosslink after the UV isomerization to form the gels.
On the contrary, no chemical reaction is necessary in our materials, but there is an electrostatic interaction between the positively charged divalent monomers and two negatively charged alginate chains. This easy gel fabrication is a significant improvement.
- In “Reversible photodissipation of composite photochromic azobenzene-alginate supramolecular hydrogels” by Anna-Lena Leistner et al., the reported bifunctional photochromic low-MW-hydrogelator contains an azobenzene core with light-dependent polarity, hydrogen-bonding cyclic dipeptide (DKP) motifs, and alkylamine side chains, whose synthesis involves eight steps and expensive protection groups reagents.
On the other hand, the cross-linking monomers used in our materials are easy to synthesize and therefore have the advantage of being inexpensive and easily fabricated, which is a desirable requirement for using these materials in real systems.
- The third reference, “Alginate-Based Smart Materials and Their Application: Recent Advances and Perspectives” by Chandan Maity & Nikita Das, is a review which we cited because it reports a comprehensive overview of the last years' innovations in alginate-based materials.
We have rephrased the introduction and several parts of the manuscript to highlight the innovative aspects and to clarify the potential applications of the materials produced in our work.
We are confident that the manuscript is now improved and that every aspect concerning the applicability and novelty of the materials presented is clearer.
Round 2
Reviewer 1 Report
Comments and Suggestions for Authors
The authors completely answered to all my comments and paper can be accepted in its present form.
Comments on the Quality of English LanguageMinor editing of English language required.
Author Response
We thank the Reviewer for the positive remarks. We have performed the required minor editing of English language.
Reviewer 3 Report
Comments and Suggestions for Authors
Thank you for revising your manuscript. The author has responded to all the reviewer's comments.
Author Response
We thank the Reviewer for the positive remarks.